# Phytochemical Screening and Antibacterial Activity of *Taxus baccata* L. against *Pectobacterium* spp. and *Dickeya chrysanthemi*

Eva Sánchez-Hernández [1], Vicente González-García [2], Jesús Martín-Gil [1], Belén Lorenzo-Vidal [3], Ana Palacio-Bielsa [2] and Pablo Martín-Ramos [1,*]

1   Department of Agricultural and Forestry Engineering, ETSIIAA, Universidad de Valladolid, Avenida de Madrid 44, 34004 Palencia, Spain
2   Department of Agricultural, Forest and Environmental Systems, Agrifood Research and Technology Centre of Aragón, Instituto Agroalimentario de Aragón—IA2 (Universidad de Zaragoza-CITA), Avda. Montañana 930, 50059 Zaragoza, Spain
3   Servicio de Microbiología, Hospital Universitario Río Hortega, Calle Dulzaina 2, 47012 Valladolid, Spain
*   Correspondence: pmr@uva.es

**Abstract:** The yew tree (*Taxus baccata* L.) is considered in folklore a symbol of immortality due to its qualities of longevity and regeneration. Despite its poisonous reputation, the yew tree has a long history of medicinal use, particularly in the form of extracts from its leaves and bark. In the work presented herein, gas chromatography–mass spectrometry (GC–MS) chemical profiling was applied to the aqueous ammonia/hydromethanolic extracts of several plant organs of *T. baccata*, leading to the identification of different bioactive compounds than those previously characterized by high-performance liquid chromatography with tandem mass spectrometry (HPLC–MS/MS) in other extraction media. The leaf aqueous ammonia extract was rich in 2-hexylthiophene and 3-*O*-methyl-D-fructose; 9-octadecenoic and hexadecanoic acid were the main constituents of the bark aqueous ammonia extract; and the fruit hydromethanolic extract contained methyl 2-*O*-methyl-$\alpha$-D-xylofuranoside, 1,3-dioxolane derivatives, and erysimoside. The antimicrobial activity of the extracts was assayed against four bacterial pathogens responsible for the soft rot and blackleg diseases of potatoes, viz. *Pectobacterium carotovorum* subsp. *carotovorum*, *Pectobacterium atrosepticum*, *Pectobacterium parmentieri*, and *Dickeya chrysanthemi*, resulting in minimum inhibitory concentration (MIC) values as low as 187 µg·mL$^{-1}$. Bioassays on potato slices confirmed the efficacy of the leaf extract at this dose when applied as a preventive treatment before artificial inoculation with *P. carotovorum* subsp. *carotovorum*. In view of this high activity, these extracts may find application in the integrated pest management of soft rot *Pectobacteriaceae* (SRP) diseases.

**Keywords:** bioactive compounds; blackleg disease; GC–MS; integrated pest management; potato; soft rot *Pectobacteriaceae*; yew tree



## 1. Introduction

The yew (*Taxus baccata* L.) is an evergreen coniferous tree in the *Taxaceae* family that is native to Western, Central, and Southern Europe, Northwestern Africa, and Southwest Asia. It grows in mixed forests, preferentially in shady areas, and is indifferent about its substrate. It is mainly grown for ornamental purposes. Yews are small to medium-sized trees that grow 10–20 m tall and have trunks up to 2 m (rarely 4 m) in diameter. The bark is thin and scaly brown, and it peels off in small flakes parallel to the stem. The leaves are flat, dark green, 1–4 cm long, and 2–3 mm wide. They are arranged in a spiral on the stem, although the leaf bases are twisted to line the leaves in two flat rows on one and the other side of the stem. The seeds are 6–7 mm long and have a woody episperm that is mostly

covered by a fleshy red aril. The yew is dioecious, so the male and female flowers grow on different trees.

The yew tree has a long history of medicinal use, particularly in the form of extracts from its leaves and bark. Concerning its medicinal uses in folklore, the leaves and stem are anti-inflammatory, antirheumatic, abortifacient, purgative, and diaphoretic, and were traditionally used for epilepsy, malaria, rheumatism, tuberculosis, cold, and coughs; a tincture from young shoots was used to treat headaches, diarrhea, biliousness, and giddiness; the bark was used to treat coughs and colds; and pounded leaves were administered orally for indigestion, asthma, and bronchial problems [1]. Nonetheless, except for the red arils, scientific data indicate that the plant's remaining parts are poisonous, including toxins that can be absorbed through skin contact and inhalation. The consumption of even a small amount of leaves, seeds, or bark can result in death. Selected compounds, such as baccatin III; 10-deacetylbaccatin III (10-DAB III); 3,5-dimethoxyphenol; paclitaxel (taxol A); cephalomannine (taxol B); 10-deacetylpaclitaxel; 10-deacetylcephalomannine; taxcultine; epicanadensene; and taxinine M were identified by high-performance liquid chromatography with tandem mass spectrometry (HPLC–MS/MS) method [2–4]. All are very toxic [2]. However, mature ripe fruits are free of toxic compounds [5–8].

Yew tree extracts have garnered significant attention as a potential treatment for cancer, given that taxol and related taxanes (e.g., docetaxel or Taxotere®) have been shown to inhibit cancer cell growth and trigger apoptosis in a variety of cancer types [9]. However, in spite of the fact that *T. baccata* is quite resistant to decay as well as to attacks of parasites and pests, pointing to the existence of bioactive compounds with potential applications in crop protection, its extracts have only been assayed against the red spider mite (*Tetranychus urticae* Koch) [10].

Concerning crop protection, a particularly interesting application would be the control of tuber soft rot caused by *Pectobacteriaceae* (SRP), which is among the most important potato (*Solanum tuberosum* L.) diseases, both for seed and ware potato production, resulting in overall losses estimated at 46 M euro/year for the European sector (with high variability among years) [11], because of the absence of authorized and effective crop protection chemicals against SRP [12], and, at least for now, lack of disease resistance. SRP can result in severe losses in important economical crops in many nations. It affects a wide host range, including three of the four most important crops in the world (viz. rice, maize, and potato).

The most common sign of potato blackleg is a slimy, moist, black rot lesion that spreads up the stems from the rotting mother tuber, particularly under wet conditions. Under dry conditions, symptoms include the stunting, wilting, yellowing, and desiccation of the leaves and stems [13]. Tuber soft rot begins at the stolon end, lenticels, and/or in wounds. Blanking happens when seed tubers start rotting in the field prior to emergence. The tuber tissue is macerated to a creamy consistency; in the presence of air, it turns black and has an unpleasant odor when secondary organisms invade it. Rotting can spread to neighboring tubers in poorly ventilated cool storage [14].

The aim of the study presented herein was twofold: (i) to investigate the constituents of needle and bark aqueous ammonia extracts, as well as of a red aril hydromethanolic extract, using gas chromatography–mass spectrometry (GC−MS) in order to analyze the influence of the extraction media and quantification methods on the presence/absence of the aforementioned phytochemicals; and (ii) to assay the antimicrobial activity of the extracts against four potato SRP species ranked in the top ten plant pathogens in 2010 [15], viz. *Pectobacterium carotovorum* subsp. *carotovorum* (Jones 1901) Hauben et al. 1999; *Pectobacterium atrosepticum* (van Hall 1902) Gardan et al. 2003; *Pectobacterium parmentieri* Khayi et al. 2016; and *Dickeya chrysanthemi* (Burkholder et al. 1953) Samson et al. 2005. The reported findings may also be applied to other vegetable crops, such as chicory and radish, which have also suffered severe soft rot epidemics in the past decades.

## 2. Material and Methods

### 2.1. Plant Material and Reagents

Samples of needle leaves, bark, and mature fruits were collected from a centenarian yew (female tree) growing in an ancient square (the *Viejo Coso*, i.e., Old Bullring [tr.]; 41°39′23.7″ N 4°43′42.0″ W) in Valladolid, Spain. Sampling took place on 20 September 2022.

Ammonium hydroxide, 50% *v/v* aq. soln. (CAS 1336-21-6), acetic acid (CAS 64-19-7; glacial, ≥99%), tryptic soy agar (TSA, CAS 91079-40-2), and tryptic soy broth (TSB, CAS 8013-01-2) were purchased from Sigma-Aldrich Química (Madrid, Spain).

### 2.2. Bacterial Strains

*Pectobacterium carotovorum* subsp. *carotovorum* (strain CITA Ecc-8), *Pectobacterium atrosepticum* (strain CITA Eca-2), and *Dickeya chrysanthemi* (strain CIRM-CFBP 7086) were supplied by the Bacteriology Laboratory at the Center for Research and Agrifood Technology of Aragón (CITA, Zaragoza, Spain). *Pectobacterium parmentieri* (strain CRD 16/111) was obtained from the Regional Diagnostic Center of the Junta de Castilla y León (Salamanca, Spain). All of these bacterial strains were isolated from diseased potatoes in Spain. The bacterial strains were cultured on TSA medium.

### 2.3. Extracts Preparation

Concerning the choice of the extraction media, the hydroalcoholic medium—a very popular option when characterization is to be conducted by GC−MS [16]—was chosen for the arils. However, it does not allow for the dissolution of polyphenols and other bioactive compounds of interest contained in the bark samples, which can be successfully attained by digestion in an aqueous ammonia solution (widely used in lignocellulosic residue pretreatment [17]), as previously shown in other works involving bark extracts [18–20]. Concerning the leaf extraction medium, aqueous ammonia was chosen due to the substantially higher extraction yield in comparison with the hydromethanolic medium (an order of magnitude higher), probably due to the high lignin content (avg. value of 18.28 ± 5.55%, according to [21]) of yew needles.

The preparation of the bark and leaf extracts was performed according to the procedure described in [22] with the modifications indicated in [20]. In brief, the powdered samples (31 g) were digested in an aqueous ammonia solution (120 mL $H_2O$ + 30 mL $NH_3$(aq.)) for 2 h before being sonicated for 10 min in pulsed mode (with a 2 min stop every 2.5 min) with a model UIP1000hdT probe-type ultrasonicator (Hielscher Ultrasonics, Teltow, Germany), neutralized with acetic acid, and allowed to stand for 24 h. After centrifuging the solution at 9000 rpm for 15 min, the supernatants were filtered using Whatman No. 1 paper. The preparation of the fruit (red arils) hydroalcoholic extract followed the same procedure, albeit in a hydromethanolic medium (100 g in 250 mL of MeOH:$H_2O$, 1:1 *v/v*) and without the neutralization step.

The samples of the extracts were freeze-dried to obtain the solid residue. The extraction yields were 1.92%, 11.15%, and 11.3% for the bark, leaf, and fruit extracts, respectively. For subsequent GC−MS analysis, 25 mg of the obtained freeze-dried extracts were dissolved in 5 mL of HPLC-grade methanol to acquire a 5 mg·mL$^{-1}$ solution, which was then filtered.

### 2.4. Extracts Characterization

The infrared spectra of the dried plant organs and their freeze-dried extracts were recorded using a Nicolet iS50 Fourier-transform infrared spectrometer (Thermo Scientific; Waltham, MA, USA) with an in-built diamond attenuated total reflectance system. The spectra were registered spanning the 400–4000 cm$^{-1}$ range, with a spectral resolution of 1 cm$^{-1}$, using the interferograms produced by co-adding 64 scans.

The extracts were further studied by GC−MS at Universidad de Alicante's Research Support Services (STI) using a model 7890A gas chromatograph coupled to a model 5975C quadrupole mass spectrometer (Agilent Technologies; Santa Clara, CA, USA). The following chromatographic conditions were used: 3 injections/vial, injection volume =

1 μL; injector temperature = 280 °C, in splitless mode; initial oven temperature = 60 °C, after 2 min, followed by a 10 °C·min$^{-1}$ increase up to a final temperature of 300 °C after 15 min. The chromatographic column utilized for compound separation was an HP-5MS UI (Agilent Technologies) of 30 m in length, 0.250 mm in diameter, and with 0.25 μm film. The mass spectrometer was configured as follows: temperature of the mass spectrometer's electron impact source = 230 °C; temperature of the quadrupole = 150 °C; ionization energy = 70 eV. For equipment calibration, test mixture 2 for apolar capillary columns according to Grob (https://www.sigmaaldrich.com/ES/es/product/sial/86501, accessed on 29 January 2023) and PFTBA were utilized as tuning standards. The components were identified by comparing their mass spectra and retention times with those of authentic compounds, as well as by computer matching with the National Institute of Standards and Technology (NIST, Gaithersburg, MD, USA) database.

### 2.5. In Vitro Antimicrobial Activity Assessment

The antibacterial activity of the extracts was assessed using the agar dilution technique in accordance with the Clinical and Laboratory Standards Institute (CLSI, Wayne, PA, USA) standard M07–11 [23], determining the minimum inhibitory concentrations (MIC). Pure cultures of each *Pectobacterium* and *Dickeya* species were incubated in a TSB liquid medium for 24 h at 28 °C. Following that, serial dilutions were performed, beginning with a concentration of $10^8$ CFU·mL$^{-1}$ and ending with a final inoculum of ~$10^4$ CFU·mL$^{-1}$. The bacterial suspensions were then spread on the surface of the TSA plates, to which the bioactive products had previously been added at concentrations in the 62.5 to 1500 μg·mL$^{-1}$ range. After 24 h of incubation at 28 °C, bacterial growth was recorded. The MICs were determined as the lowest bioactive product concentrations at which no bacterial growth was observed. All experiments were run in triplicate, with three plates for each treatment and concentration combination.

For comparison purposes, MIC values for five conventional antibiotics of clinical use (viz. benzylpenicillin or penicillin G (PG); ampicillin (AM); gentamicin (GM); ciprofloxacin (CI); and tetracycline (TC)) were determined using ETEST® strips.

### 2.6. Protection Tests on Potato Slices

The potato tubers (*Solanum tuberosum* cv. Kennebec) used to determine the protective and curative effect of *T. baccata* leaf extract were supplied by M. Prado Mazaira S.L. (Coristanco, A Coruña, Spain) and were previously cultivated according to organic farming regulations, with no use of synthetic pesticides, under the 'Pataca de Galicia' protected geographical indication. The tubers were harvested and immediately cold shipped by an express courier service so that experiments could be started within 24 h after harvesting. Tubers were selected based on uniformity of size, with a caliber ≥45 mm and an absence of physical damage and fungal or bacterial infection symptoms.

The efficacy of the treatment was determined by artificially inoculating potato slices in controlled laboratory conditions. Inoculation was carried out in accordance with the method proposed by Abd-El-Khair et al. [24]. Briefly, the potato tubers were washed under running water to remove soil and other organic particles before being superficially disinfected for 2 min with a NaOCl 3% solution, washed three times with sterile distilled water, and dried in a laminar flow hood on sterile absorbent paper. Then, under sterile conditions, potato tubers were cut into 8 mm-thick slices with a sterile knife. In each Petri plate containing sterile filter paper soaked with 3 mL of sterile distilled water, one potato slice was placed. Then, a superficial wound (ø = 3 mm) was created in the equatorial zone of each slice, where 50 μL of the bacterial suspension ($10^4$ CFU·mL$^{-1}$) of *P. carotovorum* subsp. *carotovorum* strain CITA Ecc-8 was inoculated simultaneously with the addition of 50 μL of *T. baccata* leaf extract at a concentration of 187.5 μg·mL$^{-1}$ (i.e., the MIC obtained in previous in vitro assays). The same procedure was repeated, but successively, applying the extract either 2 h before or after pathogen inoculation. The inoculated potato slices were incubated at 30 °C ± 2 for 48 h. For negative controls, potato slices were treated with distilled water.

Three independent trials were carried out, and each trial consisted of three replicate potato slices per treatment, in agreement with Garge and Nerurkar [25]. The macerated tissue was removed and weighed in order to calculate the relative percentage maceration of the potato slices, using the positive control as 100%, according to the procedure described in [26]. The results of the relative percentage maceration were statistically analyzed in IBM SPSS Statistics v.25 software (IBM, Armonk, NY, USA) by one-way analysis of variance, followed by a post hoc comparison of means by Tukey's test (because the requirements of homogeneity and homoscedasticity were met, according to the Shapiro–Wilk and Levene tests).

## 3. Results

### 3.1. Vibrational Characterization

The main bands in the infrared spectra of the different plant organs and their extracts are summarized in Table 1. The presence of thiophene rings would be supported by the aromatic C−C ring stretching at 1652 and 1374 cm$^{-1}$ and the peak observed at 720 cm$^{-1}$ for C−S bending. The bands at ca. 1460, 1245, 790, and 779 cm$^{-1}$ show the connectivity of thiophene rings through 2,5 coupling [27]. The bands at 2849, 1409, 1255, and 1193 cm$^{-1}$ have been previously reported for methyl D-lyxopyranosides [28]; those at 1143, 1023, 918, and 865 cm$^{-1}$ for dioxolanes [29]; and those at 2915, 1698, and 1102 cm$^{-1}$ for hydroxy-cyclopentenones [30].

**Table 1.** Main bands in the infrared spectra of various plant organs of *Taxus baccata* and their freeze-dried extracts, together with their assignments.

| Leaf | Leaf Extract | Bark | Bark Extract | Red Aril | Red Aril Juice | Seed | Fruit Extract | Assignment |
|---|---|---|---|---|---|---|---|---|
| 3309 | 3343 | 3324 | 3177 3000 | 3331 | 3281 | 3285 3006 | 3285 | O−H str. |
| 2920 | 2915 | 2917 | 2928 | 2920 | 2931 | 2923 | 2917 | C−H str. |
| 2851 | 2848 | | | 2851 | | 2853 | 2849 | C−H str. |
| 1727 | 1728 | | 1698 | 1726 1715 | | 1744 | | C=O str. lactones OH cyclopentenones |
| 1652 | 1652 | | | | 1636 | 1634 | | C−S ring str. furanes |
| 1615 | 1605 | 1605 | | 1609 | | | 1605 | C=O lignin |
| 1540 1507 | 1516 | 1517 | 1540 1486 | | 1549 | 1532 | | COO$^-$ N−H bending dioxolanes |
| 1457 1436 | 1463 | 1423 | | | 1443 | 1456 | | thiophenes −C=C− |
| | 1418 | | 1398 | | | 1417 | 1409 | CH$_2$/CH$_3$ |
| 1313 | 1315 | 1314 | 1341 | | 1318 | 1312 | 1347 | skeleton bending C–CH and C–OH |
| | 1279 | 1271 | | | | | 1255 | C−O str. |
| 1242 | 1245 | | | | 1242 | 1235 | | ring vib. thiophenes/xylopyranosides |
| 1159 | 1149 | 1143 | 1153 | 1187 | 1146 | 1143 | 1193 1143 | pyranose CO antisym. C−O−C |
| 1098 | 1101 | | | | 1104 | | | hydroxy-cyclopentenones |
| | 1053 | 1047 | | | 1053 | 1053 | | C−O, flavonoids |
| 1028 | 1028 | 1031 | 1015 925 | 1024 | 1031 | | 1023 918 | furanes monosaccharides |

**Table 1.** *Cont.*

| Leaf | Leaf Extract | Bark | Bark Extract | Red Aril | Red Aril Juice | Seed | Fruit Extract | Assignment |
|---|---|---|---|---|---|---|---|---|
|  | 870 | 896 | 885 |  |  | 852 | 865 | α-glucopyranoses |
| 1372 | 1374 | 1369 |  | 1373 | 1369 | 1378 |  | C−S ring str. and C−C ring aromatics |
| 836 | 822 | 808 |  |  |  |  | 817 | pyranose ring br. |
| 798 | 790 |  |  | 796 |  |  |  | C−H deform. (connect. thiophenes) |
| 779 |  |  |  | 773 |  |  | 775 | C–H β out-of-*p* bending thiophenes |
| 721 | 719 |  |  |  |  |  |  | C−S bend thiophene |

### 3.2. GC–MS Chromatograms

The main constituents identified in the leaf aqueous ammonia extract (Figures 1 and S1, Table S1) were: 2-hexylthiophene (22.4%); 3-*O*-methyl-D-fructose (16.1%); catechol (6.2%); 2,3-dimethyl-cyclohexa−1,3-diene (7.7%); phenol (6.9%) and phenol derivatives (5.8%); 2-methyl−1,3-cyclohexanedione (4.1%); 2-cyclopenten-1-one derivatives (4.1%); and 2,3-dihydro-benzofurane (3.4%).

In the bark aqueous ammonia extract (Figures 1 and S2, Table S2), the following phytochemicals were detected: 9-octadecenoic acid, methyl ester (15.7%); hexadecanoic acid, methyl ester (13.5%); undecane (12.2%); 4-acetamidobenzofuroxane (11.7%); methoxy-phenyl-oxime (11.3%); 3,4-methylenedioxyphenyl acetone (8.1%); catechol (6.4%); methyl stearate (4.8%); 1-methyl-4-(1-methylpropyl)-benzene (4.2%); dodecanoic acid methyl ester (3.5%); and 1-cyclohexene-1-carboxylic acid (3.2%).

In turn, in the fruit hydromethanolic extract (Figures 1 and S3, Table S3), GC–MS pointed to the presence of 5-hydroxymethylfurfural (5-HMF, 13.3%) and furfural (1.1%); methyl 2-*O*-methyl-α-D-xylofuranoside (8.3%); 1,3-dioxolane derivatives (8.2%); 3-deoxy-D-mannoic lactone (5.9%); 2,3-dihydro-3,5-dihydroxy-6-methyl-4H-pyran-4-one (5.5%); 3-[(4-nitrobutyl)thio]−1-propene (5.3%); dihydroxyacetone (3.3%); 2,2-dimethyl-3-oxo-butanoic acid methyl ester (2.4%); 1,4-dinitroso piperazine (1.6%); 2,3-dihydroxy propanal; 2,4-dihydroxy-2,5-dimethyl-3(2H)-furan-3-one (1.5%); 2-ethyltetrahydro-thiophene (1.3%); erysimoside or (3*β*,5*β*)-3-{[2,6-dideoxy-4-*O*-(*β*-D-glucopyranosyl)-*β*-D-ribo-hexopyranosyl]oxy}-5,14-dihydroxy−19-oxocard-20(22)-enolide (1.0%); 2-hydroxy-2-cyclopenten-1-one (0.7%); tetrahydro-2-methylthiophene (0.9%); 2-hexyl thiophene (0.8%); and catechol (0.3%).

### 3.3. Antibacterial Activity of the Extracts

#### 3.3.1. In vitro Assays

Table 2 summarizes the antibacterial activity results against the four SRP taxa. The hydromethanolic fruit extract was the least effective in all cases, with MIC values of 1000–1500 μg·mL$^{-1}$; the bark aqueous ammonia extract showed an intermediate activity, with MIC values ranging from 500 to 750 μg·mL$^{-1}$; whereas the leaf aqueous ammonia extract resulted in the highest inhibition, with MIC values of 187.5 μg·mL$^{-1}$ against all bacterial strains.

MIC values for five conventional antibiotics (viz. benzylpenicillin, ampicillin, gentamicin, ciprofloxacin, and tetracycline), determined using ETEST® strips (Figure S4), are provided in Table 3 for comparison purposes.

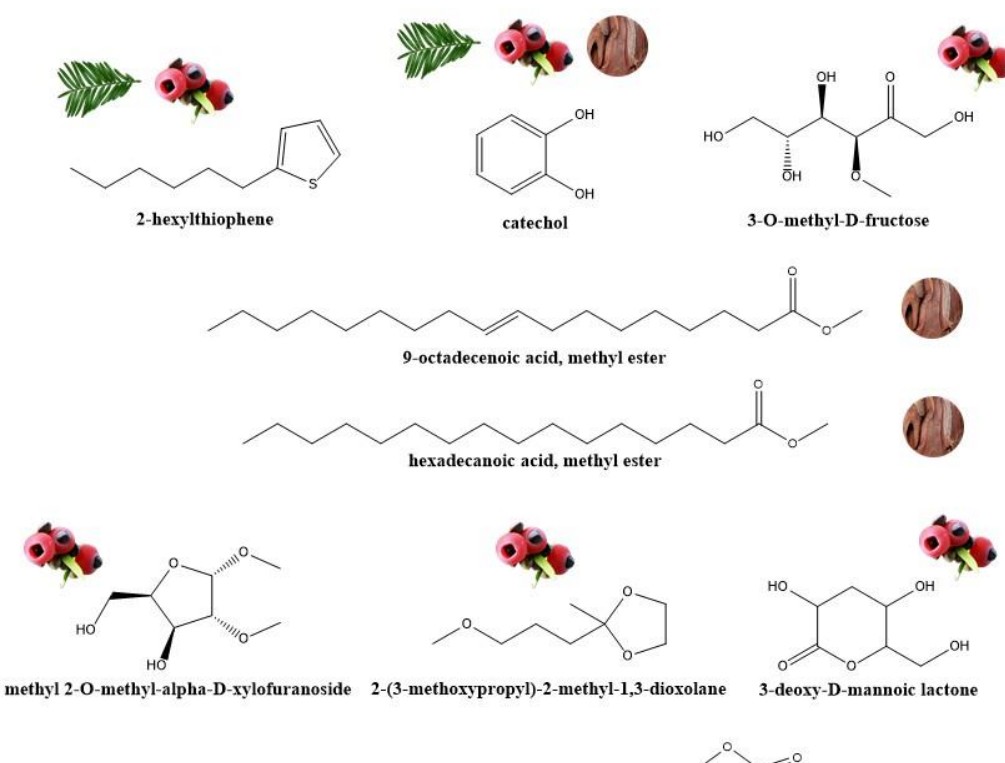

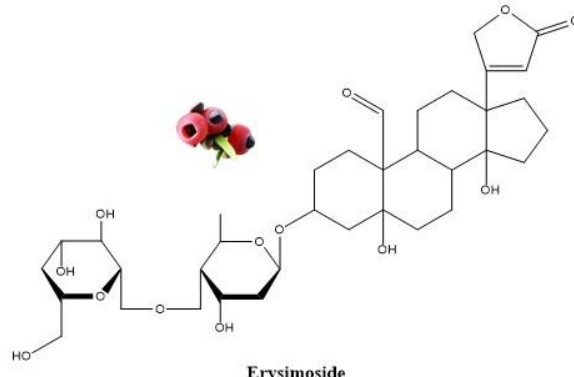

**Figure 1.** Main phytochemicals identified in *Taxus baccata* extracts by GC–MS. Pictograms indicate the organ of the plant in whose extract the compound is present.

**Table 2.** Minimum inhibitory concentrations of *Taxus baccata* extracts against several soft rot *Pectobacteriaceae* (SRP) taxa, expressed in µg·mL$^{-1}$.

| Bacteria | Fruit Extract | Bark Extract | Leaf Extract |
|---|---|---|---|
| *P. carotovorum* subsp. *carotovorum* | 1500 | 500 | 187.5 |
| *P. atrosepticum* | 1500 | 500 | 187.5 |
| *P. parmentieri* | 1000 | 500 | 187.5 |
| *D. chrysanthemi* | 1500 | 750 | 187.5 |

### 3.3.2. Bioassays on Potato Slices

Figure 2 visually displays the effects of applying the *T. baccata* aqueous ammonia leaf extract at a concentration of 187.5 µg·mL$^{-1}$ (i.e., the MIC value obtained in the in vitro tests) on the presence of soft rot disease in potato slices artificially inoculated with *P. carotovorum* subsp. *carotovorum* strain CITA Ecc-8. The corresponding relative maceration percentages are summarized in Table 4. Differences were observed depending on whether the extract was applied at the time of pathogen inoculation (Figure 2c), 2 h after the inoculation (Figure 2d), or 2 h before inoculation (Figure 2e). After 48 h, no protective effect was

observed when the extract was applied simultaneously with the pathogen or when the pathogen had been inoculated first, with percentages of maceration of 88 ± 1.4% and 70 ± 4.3%, slightly lower than those of the positive control (97 ± 3.5%, Figure 2b). In all of them, the potato slices oozed a putrid viscous liquid with a foul odor. Conversely, when the extract was applied 2 h before the inoculation (Figure 2e), it exerted a clear protective effect, controlling the incidence of soft rot disease on the potato slices, which only showed a change in color resulting from the oxidation of starch similar to the one observed in the negative control slices (Figure 2a), but with no tissue maceration and lysis.

**Table 3.** Minimum inhibitory concentrations of conventional antibiotics (for clinical use) against soft rot *Pectobacteriaceae* (SRP) (expressed in $\mu g \cdot mL^{-1}$).

| Bacteria | PG | AM | GM | CI | TC |
|----------|-----|------|------|--------|------|
| *P. carotovorum* subsp. *carotovorum* | 0.38 | 0.19 | 0.75 | 0.003 | 1 |
| *P. atrosepticum* | 6 | 0.5 | 0.75 | 0.004 | 0.75 |
| *P. parmentieri* | 0.38 | 0.094 | 0.25 | 0.006 | 0.75 |
| *D. chrysanthemi* | 12 | 0.125 | 0.5 | <0.002 | 0.25 |

PG = benzylpenicillin or penicillin G; AM = ampicillin; GM = gentamicin; CI = ciprofloxacin; TC = tetracycline.

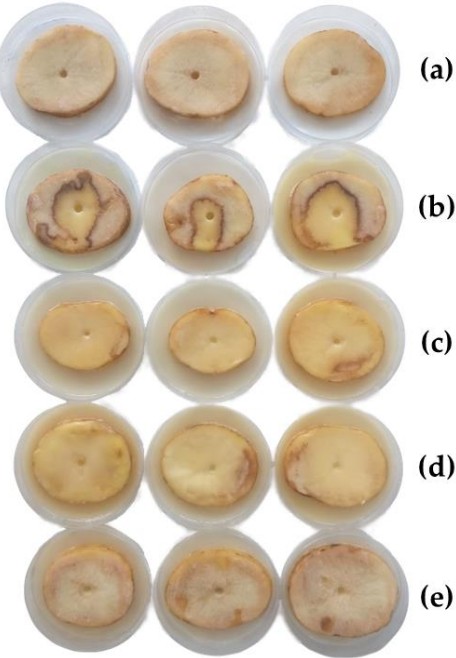

**Figure 2.** Ex situ inhibitory activity of aqueous ammonia leaf extract of *T. baccata* against *Pectobacterium carotovorum* subsp. *carotovorum* strain CITA Ecc-8 artificially inoculated on potato tuber slices: (**a**) negative control (sterile water); (**b**) positive control (pathogen and no treatment); (**c**) simultaneous inoculation of the pathogen and application of the extract; (**d**) inoculation of the pathogen and subsequent application of the extract after 2 h; and (**e**) application of the extract and subsequent inoculation of the pathogen after 2 h.

**Table 4.** Percentage of maceration of potato slices artificially inoculated with *Pectobacterium carotovorum* subsp. *carotovorum* strain CITA Ecc-8 as a function of *T. baccata* leaf extract (187.5 $\mu g \cdot mL^{-1}$) treatment application procedure.

| Treatment | Relative Maceration (%) |
|-----------|-------------------------|
| Negative control (sterile water) | 0.0 ± 0.0 % [d] |
| Positive control (no treatment) | 97 ± 3.5 % [a] |
| Simultaneous inoculation and treatment application | 88 ± 1.4 % [b] |

**Table 4.** *Cont.*

| Treatment | Relative Maceration (%) |
|---|---|
| Pathogen inoculation 2 h before treatment application | 70 ± 4.3 % [c] |
| Treatment application 2 h before pathogen inoculation | 0.0 ± 0.0 % [d] |

The same letters next to relative maceration percentages indicate that they are not significantly different at $p < 0.05$.

## 4. Discussion

### 4.1. Extract Components Identification

Given that only a small subset of known organic compounds (amenable for GC−MS) is present in the largest mass spectral databases, limitations in the identification of some of the compounds present in the extracts were detected (mostly in the leaf and fruit extracts), with quality of resemblance (Qual) values—detailed in Tables S1–S3—below 80. Hence, a word of caution seems necessary: the identification of such compounds could have some value, but it could also be off. It should also be taken into consideration that the assigned Qual value depends on the database version used for the identification and the acquisition software, so the suggested Qual = 80 threshold is a way of guidance.

Concerning the bark aqueous ammonia extract, all except for two assignments had Qual values higher than 80. One of the two compounds with a lower probability would be a benzoic acid derivative, which would be consistent with the presence of benzoic acid (identified with Qual = 93). The other, identified as 3,4-methylenedioxyphenyl acetone (Qual = 72), could be (2E)-2-octenoic acid methyl ester, according to Adams [31].

In the leaf aqueous ammonia extract, good matches (with Qual values ≥ 80) were obtained for phytochemicals such as catechol, phenol and phenol derivatives, and 2-cyclopenten-1-one derivatives, whereas the Qual values for 2-hexylthiophene and 3-*O*-methyl-D-fructose would be lower (Qual = 43–35, and 50, respectively). It is worth noting that the Qual value for 2-hexylthiophene (despite its abundance and that the presence of thiophenes is strongly supported by FTIR data) may be affected by the presence of two peaks for the same compound with close RT values, attributable to small matrix-induced retention shifts (given that no derivatization was conducted).

Regarding the fruit hydromethanolic extract, the probability of the identification of phytochemicals such as erysimoside, 5-HMF, and 2,3-dihydro-3,5-dihydroxy-6-methyl-4H-pyran-4-one was high (Qual ≥ 80), whereas those of other main constituents (e.g., 3-deoxy-D-mannoic lactone, Qual = 43–74) would be affected by the aforementioned presence of multiple peaks with close retention times. The presence of the latter, as well as that of 1,3-dioxolane derivatives, would, however, be supported by FTIR data.

### 4.2. Phytochemical Profile

The main characteristic of the phytochemicals identified for *T. baccata* is their high variability as a function of the chosen extraction medium and the organ of the plant considered. For instance, the presence of thiophenes (which, according to infrared data, is common to leaves and fruits) has been more evident in the aqueous ammonia medium (22.4% in the leaf extract vs. 2.9% in the fruit extract), whereas strophantic glycosides have only been found in the hydroalcoholic medium (i.e., in the fruit extract). However, all extracts shared the presence of catechol (albeit in different percentages: 6.2, 6.4, and 0.3% for leaf, bark, and fruit extracts, respectively).

Another characteristic feature of the identified compounds is that they exhibit remarkable bioactivities. In relation to the aforementioned presence of thiophenes, numerous plant species belonging to the family *Asteraceae* produce them. These metabolites possess antimicrobial, antiviral, anti-inflammatory, larvicidal, antioxidant, insecticidal, cytotoxic, and nematicidal properties [32]. Thus, 2-hexylthiophene (previously found, for instance, in *Akebia trifoliata* (Thunb.) Koidz. [33]), 2-ethyltetrahydrothiophene, and tetrahydro-2-methyl thiophene (formerly reported in *Nerium oleander* L. and *Uncaria tomentosa* (Willd. ex Schult.) DC. [20]), identified in the extracts presented herein, may have the antimicrobial

activity preconized for other thiophene derivatives [27,34,35]. It has been reported that thiophene acetylenes (such as 10,11-erythro-xanthopappin D) possess bioactivity against P. *carotovorum*, with an MIC of 7.25 µg·mL$^{-1}$ [36].

As for other phytochemicals present in the leaf extract, 3-*O*-methyl-D-fructose has been previously identified in *Ichnocarpus frutescens* (L.) W.T.Aiton [37] and in *Clinacanthus nutans* (Burm. f.) Lindau [38], and antibacterial activity has been described by Ghosh et al. [39]. Catechol has been reported to have an antimicrobial effect on fungal pathogens such as *Fusarium oxysporum* Schltdl. and *Penicillium italicum* Wehmer [40]. 2-hydroxy-2-cyclopenten-1-one was found in a higher percentage than in *Rubia tinctorum* L. [41], *Hibiscus syriacus* L. [42], and *Scutellaria orientalis* subsp. *bornmuelleri* (Hsskn. ex Bornm.) J.R.Edm. 1980 [43]. This phytochemical exhibits antimicrobial, anti-inflammatory, and anticancer properties [44].

Concerning the constituents of the bark extract, octadecenoic acid methyl ester has been identified, for instance, in *Albizia adianthifolia* (Schumach.) [45] and *Solena amplexicaulis* (Lam.) Gandhi extracts [46]. The former showed antibacterial activity against *Escherichia coli* (Migula 1895) Castellani & Chalmers 1919, *Pseudomonas aeruginosa* (Schroeter 1872); Migula 1900, *Bacillus subtilis* (Ehrenberg 1835) Cohn 1872, and *Staphylococcus aureus* Rosenbach 1884 (MRSA) [45]. In turn, hexadecanoic acid methyl ester, present in clove alcoholic extract, was identified by Shaaban et al. [47] as the active compound responsible for its high antimicrobial effect against clinical pathogenic bacteria (including *Staphylococcus aureus* W35, *Pseudomonas aeruginosa* D31, *Klebsiella pneumoniae* DF30, and *K. pneumoniae* B40). However, it is worth noting that methyl esters could be artifacts associated with the use of methanol as a solvent for the GC−MS analyses [48].

Regarding the chemical species identified in the fruit extract, 5-hydroxymethylfurfural—also present, for instance, in *Punica granatum* L. var. *nana* hydromethanolic extract [49]—has antibacterial properties [50], with MICs in the 40–160 µg·mL$^{-1}$ range against *Klebsiella* sp. [51]. Methyl 2-*O*-methyl-*α*-D-xylofuranoside, found as a constituent of *Alternanthera sessilis* (L.) DC. [52], was reported to exhibit antimicrobial activity. Purified biosurfactants from *Lactococcus lactis* (Lister 1873) Schleifer et al. 1986, which consist of methyl-2-O-methyl-*β*-D-xylopyranoside and octadecanoic acid, showed anti-microbial activities against methicillin-resistant *S. aureus* and *E. coli* [53]. Antibacterial and antifungal activity has also been suggested for 1,3-dioxolanes [54]. As regards the cyclic ester 3-deoxy-D-mannoic lactone, previously identified in garlic [55], in *Clerodendrum viscosum* Vent. [39], and in polyporaceous mushrooms [56], it may contribute to the antibacterial activity observed against *E. coli*; *Proteus mirabilis* Hauser 1885; *Salmonella typhi* (Schroeter 1886) Warren & Scott 1930; and *Shigella flexneri* Castellani & Chalmers 1919 reported by Shobana et al. [55]. Concerning erysimoside, it is a cardenolide glycoside that consists of strophanthidin having a *β*-D-glucopyranosyl-(1->4)-2,6-dideoxy-4-*O*-*β*-D-ribo-hexopyranosyl moiety attached at position 3. It is a 5*β*-hydroxy steroid, a 14*β*-hydroxy steroid, a 19-oxo steroid, a cardenolide glycoside, a steroid saponin, and a steroid aldehyde (Figure 3). Erysimoside is a natural product found in *Erysimum crassistylum* C. Presl. and *Erysimum leptophyllum* (M.Bieb.) Andrz., and it is functionally related to 3-hydroxycard-20(22)-enolide; 3,14,16-trihydroxycard-20(20)-enolide; strophanthidin K; ouabain (G-strophantin); and proceragenin, this latter with bacteriostatic activity [57]. Although, to the best of the authors' knowledge, there are no data on the activity of cardenolides against SRP, proceragenin has activity against members of the *Enterobacteriaceae* family, with MICs of $145 \pm 5$ µg·mL$^{-1}$ (140 and 150 µg·mL$^{-1}$ for *E. coli* and *K. pneumoniae*, respectively) [57].

Consequently, in the present study, the finding of significative activity for the assayed yew tree extracts against SRP may be attributed to the effect of the aforementioned phytochemicals, either individually or by means of a synergistic behavior.

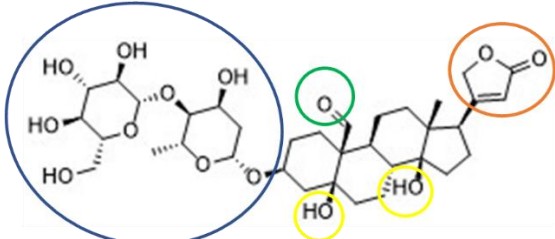

**Figure 3.** Functional groups in erysimoside: hydroxyl at 5 and 14 (in yellow); oxo at 19 (in green); β-D-glucopyranosyl-(1->4)-2,6-dideoxy-4-O-β-D-ribo-hexopyranosyl at 3 (in blue); and furanonyl (butenolide ring) at 20 (in orange).

*4.3. Comparison of Efficacy of T. baccata Extracts with Other Natural Compounds against SRP*

Tables S4–S6 compile the findings of a literature survey on the efficiency of bioactive substances of natural origin on the different SRP species studied herein. Comparisons are generally favorable for *T. baccata* extracts (except when compared against essential oils), but it must be made clear that the inhibition values listed must be interpreted with caution (even if the data pertain to the same pathogens), as the susceptibility profile varies depending on the bacterial strains, the testing methods employed, as well as the units used to express them, and also because the solvent and the utilized extraction method strongly influence the phytochemicals present in the plant extracts.

In the particular case of the comparison of activity against *P. atrosepticum* (Table S4), taking only those natural substances for which effectiveness values were expressed as MICs or effective concentrations (ECs), it may be inferred that the aqueous ammonia leaf extract of *T. baccata* (MIC = 187.5 μg·mL$^{-1}$) would be more effective than those reported for the leaf and pod of *Ceratonia siliqua* L. (MIC$_{50}$ ≤ 2400 μg·mL$^{-1}$) [58], than those of *Delonix regia* (Bojer ex Hook.) Raf. bark (MIC = 4000 μg·mL$^{-1}$) and *Erythrina humeana* Spreng. bark (MIC = 1000 μg·mL$^{-1}$) than those of the aerial part of *Eryngium triquetrum* Vahl (MIC > 40,000 μg·mL$^{-1}$), that of *Smyrnium olusatrum* L. root (MIC > 40,000 μg·mL$^{-1}$) [59], and the essential oil from *Pinus halepensis* Mill. cones (MIC = 1000–225 μg·mL$^{-1}$) [60]. Nevertheless, its activity would be slightly lower than those of the essential oils of *Cupressus macrocarpa* (Hartw. ex Gordon) Bartel branchlets (MIC = 170 μg·mL$^{-1}$) and *Corymbia citriodora* (Hook.) Hill & Johnson leaves (MIC = 170 μg·mL$^{-1}$) [61].

The activity of three yew tree extracts studied herein against *P. parmentieri* (Table S5) would also be higher than those of D. *regia* and *E. humeana* bark extracts [62], with MIC values of 4000 and 2000 μg·mL$^{-1}$, respectively.

Concerning the reported activities against *P. carotovorum* subsp. *carotovorum* (Table S6), yew tree leaf extracts (with MIC = 187.5 μg·mL$^{-1}$) would not be among the most effective natural products; in this sense, *Origanum rotundifolium* Boiss (MIC = 7.8 μg·mL$^{-1}$) [63], *Satureja khuzistanica* Jamzad (sub-MIC = 90 μg·mL$^{-1}$), *Zataria multiflora* Boiss (sub-MIC = 110 μg·mL$^{-1}$) [64], *C. macrocarpa* (MIC = 130 μg·mL$^{-1}$), and *C. citriodora* (MIC = 160 μg·mL$^{-1}$) [61] essential oils would be more active. On the other hand, it would be substantially more effective than *Hyssopus officinalis* L. essential oil (sub-MIC = 10,000 μg·mL$^{-1}$) [64] and D. *regia* and *E. humeana* bark extracts [62] (with MIC values of 2000 μg·mL$^{-1}$).

With regard to the performance of the extracts against D. *chrysanthemi* (Table S7), *T. baccata* extracts would be less active than the essential oils from *O. rotundifolium* (MIC = 7.8 μg·mL$^{-1}$) [63], *Anisomeles indica* (L.) Kuntze (MIC = 31.25 μL·mL$^{-1}$) [65] and *Salvia hians* Royle ex. Benth (31.25 μL·mL$^{-1}$) [66], and—once again—more effective than D. *regia* and *E. humeana* bark extracts [62] (with MIC values of 500 μg·mL$^{-1}$).

*4.4. Comparison with Other Extracts Tested for Potato Protection*

To the best of the authors' knowledge, there have been few studies on the use of natural extracts for protection against SRP on potato tubers. The effectiveness of *Olea europaea* L. and *C. siliqua* leaf ethanolic and ketonic extracts (at doses of 4000 and 8000 μg·mL$^{-1}$) on

their capacity to lessen the severity of soft rot was examined by Ouanas et al. [67]. The olive tree leaf ethanolic extract attained positive results, with a decrease in rot weight as compared to the control treatments, whereas the leaf ketonic extract performed just marginally better. In contrast, regardless of the quantity of the extract employed, carob's ethanolic and ketonic extracts dramatically increased the rotting process. On the other hand, the pretreatment of potato tubers with an ethanolic extract of oak bark (1000 $\mu$g·mL$^{-1}$) reduced the onset of tissue maceration symptoms, with the percentage of damaged tissue always remaining below 10% after three days of incubation [68]. Finally, the application of a 40% aqueous extract of leaves and stems of *Tagetes minuta* L. showed no rotting symptoms after 11 days of incubation, whereas the application of a 40% aqueous extract of fruits of *Capsicum frutescens* L. showed even greater rotting than the negative control [69]. The dose required herein to attain the slices' protection when applied prior to the pathogen inoculation was substantially lower than those indicated above, although further research is needed to determine if higher extract concentrations can protect the tubers when applied simultaneously or after pathogen inoculation. In addition, the results obtained here by the direct application of *T. baccata* extracts to potato slices should be confirmed when they are applied as a systemic pre-treatment on intact tubers.

*4.5. Comparison with Conventional Antibiotics*

Even though conventional antibiotics should not be used for crop protection purposes due to heightened global concern about antimicrobial resistance, three classes of antibiotics commonly used in plant production are also routinely used to treat human and animal diseases, viz. aminoglycosides, tetracyclines, and quinolones [70]. The efficacy of the tested antibiotics of these three groups (viz. gentamicin, tetracycline, and ciprofloxacin) against the four SRP taxa was orders of magnitude higher than those of the natural extracts, with MIC values below 1 $\mu$g·mL$^{-1}$ in all cases. Similar efficacy was observed for ampicillin and benzylpenicillin beta-lactam antibiotics, except against *P. atrosepticum*, in which resistance was observed for penicillin G. However, despite the lower efficacy of the natural extracts reported herein, it is worth noting that they are one of the most promising approaches to support judicious pesticide use. In fact, their use would be aligned with the World Health Organization (WHO) Global Action Plan on Antimicrobial Resistance, which considers such biorational products of low risk to the environment and human health.

**5. Conclusions**

New crop protection trends have shifted away from reliance on conventional pesticides. Consequently, the interest in effective and sustainable alternative strategies to conventional pesticides has increased. The antimicrobial activity of *T. baccata* leaves, bark, and fruit extracts, assayed against bacterial pathogens responsible for soft rot diseases of potatoes, viz. *Pectobacterium carotovorum* subsp. *carotovorum* strain Ecc-8, *P. atrosepticum* strain Eca-2, *P. parmentieri* strain CRD 16/111, and D. *chrysanthemi* strain CRD 16/111, resulted in MIC values as low as 187 $\mu$g·mL$^{-1}$, being among the lowest reported in the literature for plant extracts. The antimicrobial activity of these *T. baccata* extracts can be attributed to the presence of thiophenes and other phytoconstituents (3-*O*-methyl-D-fructose, 2-hydroxy-2-cyclopenten-1-one, octadecenoic and hexadecenoic acid methyl esters, 5-HMF, methyl 2-*O*-methyl-$\alpha$-D-xylofuranoside, 2-(3-methoxypropyl)-2-methyl$-$1,3-dioxolane, 3-deoxy-D-mannoic lactone, and erysimoside) present in the extracts. Ex situ tests conducted on potato cv. Kennebec slices confirmed the leaf extract potential for the control of artificially inoculated *P. carotovorum* subsp. *carotovorum* strain CITA Ecc-8 when applied as a preventive treatment, avoiding rotting symptoms at a dose equal to the in vitro-determined MIC. In view of this promising activity, yew tree extracts may deserve attention as an integrated pest management strategy applicable to soft rot *Pectobacteriaceae* (SRP) disease control.

**Supplementary Materials:** The following supporting information can be downloaded at: https://www.mdpi.com/article/10.3390/horticulturae9020201/s1, Figure S1. GC–MS chromatogram of *Taxus baccata* leaf aqueous ammonia extract; Figure S2. GC–MS chromatogram of *Taxus baccata* bark aqueous ammonia extract; Figure S3. GC–MS chromatogram of *Taxus baccata* fruit hydromethanolic extract; Figure S4: Examples of determination of minimum inhibitory concentrations of conventional antibiotics (for clinical use) against *P. carotovorum* subsp. *carotovorum*; *P. atrosepticum*; *P. parmentieri*; and *D. chrysanthemi* using ETEST® gradient MIC strips; Table S1. Phytochemicals identified by GC–MS in *T. baccata* leaf aqueous ammonia extract; Table S2. Main phytochemicals identified by GC–MS in *T. baccata* bark aqueous ammonia extract; Table S3. Phytochemicals identified by GC–MS in *T. baccata* fruit hydromethanolic extract; Table S4. Effectiveness values reported in the literature for natural substances against *P. atrosepticum* [58–62,71–75]; Table S5. Effectiveness values reported in the literature for natural substances against *P. parmentieri* [62,71]; Table S6. Effectiveness values reported in the literature for natural substances against *P. carotovorum* subsp. *carotovorum* [61–64,71,72,76–78]; Table S7. Effectiveness values reported in the literature for natural substances against D. *chrysanthemi* [62,63,65,66,79–83].

**Author Contributions:** Conceptualization, V.G.-G. and A.P.-B.; methodology, B.L.-V. and A.P.-B.; validation, A.P.-B.; formal analysis, E.S.-H., V.G.-G. and P.M.-R.; investigation, E.S.-H., V.G.-G., J.M.-G., B.L.-V., A.P.-B. and P.M.-R.; resources, J.M.-G. and A.P.-B.; writing—original draft preparation, E.S.-H., V.G.-G., J.M.-G., B.L.-V., A.P.-B. and P.M.-R.; writing—review and editing, E.S.-H., V.G.-G. and P.M.-R.; visualization, E.S.-H.; supervision, P.M.-R. All authors have read and agreed to the published version of the manuscript.

**Funding:** This research received no external funding.

**Data Availability Statement:** The data presented in this study are available on request from the corresponding author. The data are not publicly available due to their relevance to an ongoing Ph.D. thesis.

**Acknowledgments:** To Pilar Blasco and Pablo Candela from the Technical Research Services of the University of Alicante for conducting the GC–MS analysis. To José Luis Palomo and Jaime Alonso Herrero from the Regional Diagnostic Center of the Junta de Castilla y León (Aldearrubia, Salamanca, Spain) for providing the *P. parmentieri* strain.

**Conflicts of Interest:** The authors declare no conflict of interest.

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
