# Peer review of "Phytochemical Screening and Antibacterial Activity of Taxus baccata L. against Pectobacterium spp. and Dickeya chrysanthemi"

_horticulturae, doi:10.3390/horticulturae9020201_

Round 1

Reviewer 1 Report

It is not clear what the main goal pursued by the authors. Two blocks of results are traceable:

1) Evaluation of antimicrobial properties and component composition of extracts.

2) Prospects for the use of T. baccata extracts as a means of protecting potatoes (and other plants) from phytopathogens.

In this regard, I have two remarks concerning two aspects of this work.

      Identification of components separated by gas chromatography occurs with different probability, expressed as a percentage.  It is necessary to mention only those components that have a high percentage of homology with substances from databases >80% (paragraph 3.2, Table S1, S2). In addition, I would recommend comparing extracts from fruits, bark and leaves in terms of antimicrobial activity (MIC value) and the qualitative and quantitative content of matching chemicals.

        Another comment related to ex vivo results with potato slices. Photographs alone are not enough; it is necessary to assess the degree of tissue maceration in each variant of the experiment. How it's done in the works:  https:// doi. org/ 10. 1007/s11274- 020- 02982-4 ; https:// doi. org/10.1007/s11274-022-03366-6. The obtained quantitative results should be processed by adequate statistical criteria. The lack of information about the number of objects taken in the experiment (n= ?) and how many experiments were done in total are confusing.

 Minor remarks

Lines 56-60. Mentioned compounds are toxic or beneficial?

Line 61. Mature rip fruits are non-toxic because they are taxane-free ? What is taxane ? rewrite this sentence to bring clarity

Why did you capitalize "soft rot"?

Legend of Figure 1. It is not clear by what criterion the substances were selected.

Lines 235-236. Move this sentence into discussion section or delete

Table 2. Delete all + and – Indicate only the MIC values.

Author Response

Q1. It is not clear what the main goal pursued by the authors. Two blocks of results are traceable:

1) Evaluation of antimicrobial properties and component composition of extracts.

2) Prospects for the use of T. baccata extracts as a means of protecting potatoes (and other plants) from phytopathogens.

Response: As noted by the Reviewer, the aim of the study is twofold: (1) to investigate the yew extracts constituents; (2) to assess their antibacterial activity against SRP species. The final paragraph of the introduction has been slightly rewritten to make this twofold aim clear.

In this regard, I have two remarks concerning two aspects of this work.

Q2. Identification of components separated by gas chromatography occurs with different probability, expressed as a percentage.  It is necessary to mention only those components that have a high percentage of homology with substances from databases >80% (paragraph 3.2, Table S1, S2).

Response: To address the Reviewer’s request, we have included the NIST library search reports provided by the external laboratory to which the GC-MS analyses were outsourced. The probability of the identification (i.e., quality of resemblance) has been included in Tables S1-S3 (‘Qual’ column). We are aware of the limitations in the identification of some of the compounds (mostly in the leaf and fruit extracts), given that only a small subset of known organic compounds (amenable for GC-MS) is present in the largest mass spectral databases (such as NIST or Wiley). Moreover, the NIST version used by the lab appears to be an older version (NIST11), which includes a lot less than the later editions, and we do not know what acquisition software was used (e.g., in older versions of Chemstation, matches were low due to the library search comparing the entire mass range of its spectra with the smaller mass range of the acquired data, while Shimadzu's GC-MS solutions do not do this, so matches that would normally record 75-80% register as 95%).

We agree with the Reviewer that reporting compounds with a lower fit than 80% brings the interpretation into the grey zone: the identification could have some value, but it could also be completely off. If there was a need for accurate compound identifications for safety assessments, we would certainly avoid reporting compounds with spectral match factors lower than 80%, but -in this case- there are not. Taking into consideration that the occurrence of the main compounds is documented in other plant extracts (subsection 4.2), which would reinforce the database suggestions, and that FTIR data also supports the presence of some of the compounds with low ‘Qual’ values (such as thiophenes), we have decided to keep the original results section with the ‘best guesses’ obtained from the NIST11 database, but we have created a new subsection (4.1) in the discussion to specifically address this Reviewer’s query and make the limitations clear to the readers.

Q3. In addition, I would recommend comparing extracts from fruits, bark and leaves in terms of antimicrobial activity (MIC value) and the qualitative and quantitative content of matching chemicals.

Response: Please kindly note that the requested comparison of MIC values was already provided in Table 2 (which has been updated according to Q10 below), and was briefly discussed in the first paragraph of subsection 3.3.1. A qualitative comparison of matching chemicals was presented in the first paragraph of subsection 4.1 and in Figure 1 using pictograms, but percentages have been specified to include quantitative data for the shared compounds (viz. thiophenes and catechol).

Q4. Another comment related to ex vivo results with potato slices. Photographs alone are not enough; it is necessary to assess the degree of tissue maceration in each variant of the experiment. How it's done in the works:  https://doi.org/10.1007/s11274-020-02982-4 ; https://doi.org/10.1007/s11274-022-03366-6. The obtained quantitative results should be processed by adequate statistical criteria. The lack of information about the number of objects taken in the experiment (n= ?) and how many experiments were done in total are confusing.

Response: A new table (Table 4) has been added to show the results of the relative percentages of maceration of the potato slices, calculated according to the procedure described in one of the references suggested by the Reviewer. Results have been processed using ANOVA followed by Tukey’s HSD test. Subsection 2.6 in Materials and Methods has been updated accordingly, to include both the maceration percentages determination and the statistical analysis procedure. Please kindly note that Subsection 3.3.2 has also been updated, given a typo was detected (the claims made in the text did not match the lettering of Figure 2).

Regarding the number of potato slices used in the experiment (3 independent trials × 3 potato slices/(treatment and trial) × 5 treatments (including positive and negative controls) = 45 potato slices), we have rephrased the sentence using the original wording of the work by Garge et al. [19], i.e., “Three independent trials were carried out and each trial consisted of three replicate potato slices per treatment […]”, to avoid potential confusion.

Minor remarks

Q5. Lines 56-60. Mentioned compounds are toxic or beneficial?

Response: The mentioned compounds (baccatin III, 10-deacetylbaccatin III, 3,5-dimethoxyphenol, paclitaxel, cephalomannine, 10-deacetylpaclitaxel, 10-deacetylcephalo mannine; taxcultine, epicanadensene and taxinine M) all are very toxic, even though some of them can be used (in a similar fashion to Paclitaxel and Docetaxel commercial drugs) for cancer treatment (which was explained in the following paragraph). We have added a clarification with a supporting reference: […] All are very toxic [2]. […]”.

Q6. Line 61. Mature rip fruits are non-toxic because they are taxane-free? What is taxane ? rewrite this sentence to bring clarity

Response: The sentence was correct (taxanes are natural diterpenoids occurring in yew plants, which received their name from a Latin term for yew, Taxus, and feature a taxadiene core), but to bring clarity, we have rewritten it as follows: “[…] However, mature ripe fruits are free of toxic compounds [5-8]”

Q7. Why did you capitalize "soft rot"?

Response: Corrected throughout the manuscript.

Q8. Legend of Figure 1. It is not clear by what criterion the substances were selected.

Response: On the basis of chemical abundance. Thus, ‘Selected phytochemicals’ has been replaced with ‘Main phytochemicals’ in Figure 1 caption.

Q9. Lines 235-236. Move this sentence into discussion section or delete

Response: We assume that the Reviewer is referring to L228-229 (“It is worth noting that the methyl esters could be artifacts associated with the use of 228 methanol as a solvent for the GC−MS analyses [24]”). The sentence has been moved to the discussion section, as suggested.

Q10. Table 2. Delete all + and – Indicate only the MIC values.

Response: Table 2 has been modified according to the Reviewer’s suggestion.

Reviewer 2 Report

GENERAL COMMENTS

-I understand that this manuscript explores the presence of compounds contained in two extracts of the fruits, leaves and bark, as well as the in vitro antibacterial activity against four species of the complex responsible for the SRP. Additionally, the presence of symptoms is determined after confronting extracts with bacteria in an in vitro bioassay. The manuscript is very well presented.

-I understand that this work is framed in a thesis and that the experiments should already be on the way. In case of continuing (a posteriori) in this line, it could be taken into account that Kennebec as a biological model is tolerant to SRP, using a susceptible genotype could give more evident results in terms of protection. 

SPECIFIC SUGGESTIONS

-I suggest: i) include the quantities of T. baccata leaves, fruits and bark from which the powder for the extractions was obtained and, additionally, directly cite the original work (Charles, S.J.; Russell, G.K. Chemical products from bark digested in ammonia U.S. Patent No. 2,823,223, 11 February 1958) and not the reference [16] which says exactly the same as in this work. Reference [16] also does not indicate the initial amount of material, so this work with all the information would be a better reference for future citation.

-I think it would be good to mention the reasons why the part of the plant-solution combinations (ammonia-bark/leaves and hydroalcoholic-fruit) were chosen.

-In line 254 there is "incidence", however I think it should be "presence" considering that in phytopathology the incidence is understood as the frequency of diseases within a whole (population).

-The description of Fig. 2e in line 257 does not match that on 270. The symptoms of Fig. 2e correspond to the application 2h after the inoculation of the bacteria. In general, the result of the protection assay could be more informative if described in more concrete terms so that it is more useful when differentiating extracts. For example, consider an aditional table (or text) to the Fig. 2 in quantitative terms (eg % necrotic area) or qualitative terms (presence/absence of symptoms, type of symptom). 

OTHER DETAILS

-The abbreviation PGI is not necessary as it is not used in the rest of the document.

-In the tables of results you can use the short name of the bacteria, and without the code of the strains.

Author Response

GENERAL COMMENTS

I understand that this manuscript explores the presence of compounds contained in two extracts of the fruits, leaves and bark, as well as the in vitro antibacterial activity against four species of the complex responsible for the SRP. Additionally, the presence of symptoms is determined after confronting extracts with bacteria in an in vitro bioassay. The manuscript is very well presented.

Response: Thank you for your positive feedback.

Q1. I understand that this work is framed in a thesis and that the experiments should already be on the way. In case of continuing (a posteriori) in this line, it could be taken into account that Kennebec as a biological model is tolerant to SRP, using a susceptible genotype could give more evident results in terms of protection.

Response: Thank you very much for pointing this out. We will take it into account for future experiments.

SPECIFIC SUGGESTIONS

Q2. I suggest: i) include the quantities of T. baccata leaves, fruits and bark from which the powder for the extractions was obtained and, additionally, directly cite the original work (Charles, S.J.; Russell, G.K. Chemical products from bark digested in ammonia U.S. Patent No. 2,823,223, 11 February 1958) and not the reference [16] which says exactly the same as in this work. Reference [16] also does not indicate the initial amount of material, so this work with all the information would be a better reference for future citation.

Response: The quantities of T. baccata leaves, fruits, and bark used to prepare the extracts have been indicated (together with the extraction yields), and a reference to (expired) US patent US2823223A on ‘Chemical products from bark digested in ammonia’ has been included. Please kindly note that the original 1958 procedure had been substantially modified in reference [16], incorporating Green Chemistry techniques, so both references have been included in the revised version of the ms.

Q3. I think it would be good to mention the reasons why the part of the plant-solution combinations (ammonia-bark/leaves and hydroalcoholic-fruit) were chosen.

Response: Concerning the choice of the extraction media, the hydroalcoholic medium is a very popular option when characterization is to be conducted by GC-MS [J Pharm Bioallied Sci. 2020 Jan-Mar; 12(1): 1–10; doi: 10.4103/jpbs.JPBS_175_19] and was thus chosen for the arils. However, it does not allow for the dissolution of polyphenols and other bioactive compounds of interest contained in bark samples, which can be successfully attained by digestion in an aqueous ammonia solution, as we have previously confirmed in other works involving bark extracts [Int. J. Mol. Sci. 2022, 23(19), 11882; https://doi.org/10.3390/ijms231911882; Plants 2022, 11(24), 3415; https://doi.org/10.3390/plants11243415; Horticulturae 2022, 8(8), 672; https://doi.org/10.3390/horticulturae8080672]. In this regard, it should be noted that it is widely used in lignocellulosic residue pretreatment, because -as noted in [https://doi.org/10.1016/j.bej.2021.108106]- “aqueous ammonia pretreatment is regarded as an efficient pretreatment process since it could remove acetyl groups from xylan polymers, selectively decompose and remove lignin from substrates, decrease cellulose crystallinity and enhance porosity with low release of sugar degradation compounds. Furthermore, aqueous ammonia pretreatment is inexpensive, safe to handle, non-polluting, non-corrosive and recyclable due to its high volatility”. Concerning the needle extraction medium, aqueous ammonia was finally chosen due to the substantially higher extraction yield in comparison with the hydromethanolic medium (11.2% vs. 1.9%, respectively). This should be ascribed to the high lignin content (avg. value of 18.28±5.55, according to Petisco et al. [https://digital.csic.es/bitstream/10261/10138/1/6.pdf]) of yew needles. A clarification has been included at the beginning of subsection 2.3.

Q4. In line 254 there is "incidence", however I think it should be "presence" considering that in phytopathology the incidence is understood as the frequency of diseases within a whole (population).

Response: ‘Incidence’ has been replaced with ‘presence’, as suggested by the Reviewer.

Q5. The description of Fig. 2e in line 257 does not match that on 270. The symptoms of Fig. 2e correspond to the application 2h after the inoculation of the bacteria. In general, the result of the protection assay could be more informative if described in more concrete terms so that it is more useful when differentiating extracts. For example, consider an additional table (or text) to the Fig. 2 in quantitative terms (eg % necrotic area) or qualitative terms (presence/absence of symptoms, type of symptom).

Response: We thank the Reviewer for pointing out this mistake, which has been corrected in the revised version. Additionally, a new table (Table 4) has been added, summarizing the relative maceration percentages of the potato slices to complement with quantitative data the qualitative information provided in Figure 2, as suggested by the Reviewer. The text in materials and methods, as well as in subsection 3.3.2, has been updated accordingly.

OTHER DETAILS

Q6. The abbreviation PGI is not necessary as it is not used in the rest of the document.

Response: The abbreviation has been removed.

Q7. In the tables of results you can use the short name of the bacteria, and without the code of the strains.

Response: Corrected in Table 2 and Table 3.

Reviewer 3 Report

In this work, Sánchez-Hernández et al. present the chemical profile of the aqueous/ammonia extracts of several parts of the yew tree and the antibacterial properties of those extracts. It is a very well-written manuscript, containing a clear and thorough introduction and methodology. The results are clearly justified (except for one point - see below) and the discussion is complete.

My main remark, probably due to the misassignment of the lettering of Figure 2, is that I cannot follow the writers' claims in Lines 255-257. Please, rectify this.

Another minor issue is in Line 196 and Line 235, in which -obviously- table references are erroneous.

Author Response

In this work, Sánchez-Hernández et al. present the chemical profile of the aqueous/ammonia extracts of several parts of the yew tree and the antibacterial properties of those extracts. It is a very well-written manuscript, containing a clear and thorough introduction and methodology. The results are clearly justified (except for one point - see below) and the discussion is complete.

Q1. My main remark, probably due to the misassignment of the lettering of Figure 2, is that I cannot follow the writers' claims in Lines 255-257. Please, rectify this.

Response: We thank the Reviewer for pointing out this mistake, which has been corrected in the revised version. Additionally, a new table (Table 4) has been added, summarizing the relative maceration percentages of the potato slices to complement with quantitative data the qualitative information provided in Figure 2. The text in materials and methods and in subsection 3.3.2 has been updated accordingly.

Q2. Another minor issue is in Line 196 and Line 235, in which -obviously- table references are erroneous.

Response: The problem with the cross-references to the tables has been fixed throughout the ms. Please kindly note that minor formatting-related adjustments were made by the journal to the version of the manuscript that we originally submitted before starting the peer-review process: unfortunately, they deleted the automatic numbering of table headers and MS Word’s cross-references were no longer valid. References to tables and figures are now made as regular text to circumvent the problem.

Round 2

Reviewer 1 Report

I am satisfied with the answers of the Authors and the changes made